# Favorable Effects of Voriconazole Trough Concentrations Exceeding 1 μg/mL on Treatment Success and All-Cause Mortality: A Systematic Review and Meta-Analysis

**DOI:** 10.3390/jof7040306

**Published:** 2021-04-16

**Authors:** Yuki Hanai, Yukihiro Hamada, Toshimi Kimura, Kazuaki Matsumoto, Yoshiko Takahashi, Satoshi Fujii, Kenji Nishizawa, Yoshitsugu Miyazaki, Yoshio Takesue

**Affiliations:** 1Department of Pharmacy, Toho University Omori Medical Center, Tokyo 143-8541, Japan; yuuki.hanai@med.toho-u.ac.jp (Y.H.); nishizawa@med.toho-u.ac.jp (K.N.); 2Department of Pharmacy, Tokyo Women’s Medical University Hospital, Tokyo 162-8666, Japan; t.kimura.pha@twmu.ac.jp; 3Division of Pharmacodynamics, Keio University Faculty of Pharmacy, Tokyo 105-8512, Japan; matsumoto-kz@pha.keio.ac.jp; 4Department of Pharmacy, Hyogo College of Medicine, Hyogo 663-8501, Japan; yktabu@hyo-med.ac.jp; 5Department of Hospital Pharmacy, Sapporo Medical University Hospital, Hokkaido 060-8543, Japan; fujii.satoshi@sapmed.ac.jp; 6Department of Chemotherapy and Mycoses, National Institute of Infectious Diseases, Tokyo 162-8640, Japan; ym46@nih.go.jp; 7Department of Infection Control and Prevention, Hyogo College of Medicine, Hyogo 663-8501, Japan; takesuey@hyo-med.ac.jp

**Keywords:** voriconazole, meta-analysis, trough concentration, therapeutic drug monitoring, mortality

## Abstract

This systematic review and meta-analysis examined the optimal trough concentration of voriconazole for adult patients with invasive fungal infections. We used stepwise cutoffs of 0.5–2.0 μg/mL for efficacy and 3.0–6.0 μg/mL for safety. Studies were included if they reported the rates of all-cause mortality and/or treatment success, hepatotoxicity, and nephrotoxicity according to the trough concentration. Twenty-five studies involving 2554 patients were included. The probability of mortality was significantly decreased using a cutoff of ≥1.0 μg/mL (odds ratio (OR) = 0.34, 95% confidence interval (CI) = 0.15–0.80). Cutoffs of 0.5 (OR = 3.48, 95% CI = 1.45–8.34) and 1.0 μg/mL (OR = 3.35, 95% CI = 1.52–7.38) also increased the treatment success rate. Concerning safety, significantly higher risks of hepatotoxicity and neurotoxicity were demonstrated at higher concentrations for all cutoffs, and the highest ORs were recorded at 4.0 μg/mL (OR = 7.39, 95% CI = 3.81–14.36; OR = 5.76, 95% CI 3.14–10.57, respectively). Although further high-quality trials are needed, our findings suggest that the proper trough concentration for increasing clinical success while minimizing toxicity is 1.0–4.0 μg/mL for adult patients receiving voriconazole therapy.

## 1. Introduction

Voriconazole is a triazole with broad-spectrum activity against most clinically significant yeasts and molds [1,2,3], and it is available as both oral and intravenous formulations. Voriconazole serum concentrations are highly variable because of its non-linear pharmacokinetics, and they are further influenced by factors such as drug interaction, altered intestinal absorption, and genetic polymorphism. These pharmacokinetic variabilities have important implications for dosage adjustment because of unpredictable changes in drug exposure. Therefore, therapeutic drug monitoring (TDM) is used to guide voriconazole therapy to prevent drug-related adverse events and improve clinical responses by individualizing dose regimens [4,5].

In many institutions, monitoring has become routine for serum voriconazole concentrations, and antifungal stewardship (AFS) programs incorporate TDM in Japan [6]. To improve the outcomes of voriconazole treatment, TDM is suggested in major guidelines from the Infectious Diseases Society of America, the American Thoracic Society, and the European Society of Clinical Microbiology and Infectious Diseases [1,2,3,7,8,9]. A guideline [10] published by the Japanese Society of Chemotherapy and Japanese Society of Therapeutic Drug Monitoring in 2013 recommended a voriconazole target trough concentration of >1–2 μg/mL for efficacy and a trough concentration of <4–5 μg/mL for avoiding alterations of liver function. This guidance was primarily based on a meta-analysis of observational studies by Hamada et al. [11]. The revised version also includes the results of a pediatric meta-analysis [12].

However, data on the target range of voriconazole concentrations remains a matter of debate because most studies had several limitations. As one reason, previous studies [11,13,14] were conducted to identify the optimal trough concentration of voriconazole by meta-analysis, but the target concentrations are different. Additionally, a survival benefit is considered the most important clinical outcome for therapeutic intervention, but these studies did not clarify the relationship between mortality and sub-therapeutic trough concentrations. Furthermore, these studies had drawbacks such as the lack of inclusion of eligible studies (searched from its inception until March 2015) and inadequate population characteristics in both children and adults. To date, no randomized trials have assessed voriconazole target trough concentrations or efficacy and safety in patients with invasive fungal infections (IFIs). Therefore, the optimal trough concentrations of voriconazole against IFIs remain unclear and controversial, and an updated meta-analysis is required to provide recommendations regarding these concentrations. The present study evaluated the relationship between the reported voriconazole trough concentration and mortality as the primary endpoint, and the secondary aim was to reassess the efficacy and safety of this concentration in adults with IFIs using a systematic review and meta-analysis.

## 2. Materials and Methods

### 2.1. Search Strategy and Selection Criteria

Following the Preferred Reporting Items for Systematic Reviews and Meta-Analyses statement, we searched electronic databases (PubMed, Cochrane Library, Web of Science, ClinicalTrials.gov, and Japana Centra Revuo Medicina) for clinical studies published up to February 1, 2021 using a combined MeSH heading and text search strategy with the following terms: ‘voriconazole’, ‘vfend’, ‘drug monitoring’, and ‘pharmacovigilance’. We also manually checked the reference lists of relevant original papers and reviews, screened articles in the PubMed ‘related citations’ section, and restricted the search to human studies.

Studies were included for analysis if they assessed the relationship between the voriconazole concentration and clinical efficacy and/or safety, if all participants were adults (≥15 years old), if they were original articles (not review, editorials, research letters and protocols), and if they provided the incidence of a given outcome according to the voriconazole concentration.

We excluded studies if the data were generated from simulated patients or pharmacokinetic models rather than real patients, if the voriconazole concentration was not the trough concentration, and if the study was a case report. We further excluded studies if only the abstract was published and if the use of voriconazole was not discussed. If multiple articles were derived from the same studies and the same associated events were reported, then we included only the latest published data for our primary analysis.

### 2.2. Data Extraction

Two reviewers independently screened the titles and abstracts of the reports, and full copies of potentially suitable studies were obtained. Study information such as publication year, study location, number of participants, participants’ baseline characteristics, study duration, type of fungal infection, type and definition of outcomes (efficacy and/or safety), cutoff of the voriconazole trough concentration, and concomitant antifungal therapy was recorded on pretested standard forms. Disagreement on the specific data between two reviewers was resolved by discussion. We contacted the authors to obtain data missing from the original publication by email when required.

Some studies reported the MIC_90_ of voriconazole against clinical isolates of *Candida* and *Aspergillus* species as ranging between 0.5 and 1.0 μg/mL [15,16,17,18], whereas others found that efficacy increases when the trough concentration exceeds 2.0 μg/mL [19,20]. Accordingly, we set the stepwise cutoff for efficacy at a range of 0.5–2.0 μg/mL and examined outcomes. For safety, we set the stepwise cutoff at a range of 3.0–6.0 μg/mL and examined outcomes in a stepwise manner because some sources stated that the maximum value should be between 3.0 and 6.0 μg/mL [11,13,14,21].

For efficacy and safety, when the trough concentration was measured multiple times for each patient, we used the mean of multiple measurements, and the median was used only when the mean was not available. If there were multiple data for the same outcome in an article, only outcome data with the longest follow-up were extracted. When required, investigators were contacted to obtain data missing from the original publication.

### 2.3. Quality Assessment

The quality of the included studies was assessed by two investigators using the Risk of Bias Assessment tool for Non-randomized Studies (RoBANS) [22]. RoBANS includes criteria for judging the risk of bias for each domain. The risk of bias in a study was graded as low, high, or unclear for the following study features: selection of participants (selection bias), consideration of confounding variables (selection bias), measurement of exposure (detection bias), blinding of outcome assessment (detection bias), incomplete outcome data (attrition bias), and selective outcome reporting (reporting bias).

### 2.4. Outcomes and Definitions

The efficacy outcomes were all-cause mortality and treatment success related to treatment of confirmed or suspected IFIs. Given the known variations in the definitions of treatment success in the literature, we used the criteria from the majority of included studies to minimize heterogeneity. The safety outcomes were hepatotoxicity and neurotoxicity. Neurological adverse effects were defined as any neurological or visual symptoms including auditory and visual hallucinations, dizziness, tremor, consciousness disturbance, and visual disorders.

For each outcome, subgroup analyses according to study location for comparisons between Asian and non-Asian countries were performed. Asians, particularly Japanese, Korean, and Chinese populations, are known to have a higher proportion of CYP2C19 poor metabolizers (15–20%) than Caucasians and Africans (2–3%), which may influence the incidence of adverse events [23]. Similarly, to explore the heterogeneity among different studies, subgroup analyses were performed according to study design, diagnosis of fungal infection, fungal organisms, and concomitant antifungals.

### 2.5. Statistical Analysis

Data were analyzed using Review Manager 5.3 (Review Manager (RevMan) v. 5.3, Copenhagen: The Nordic Cochrane Centre, The Cochrane Collaboration, 2014), and forest plots were prepared. The odds ratios (ORs) and 95% confidence intervals (CIs) were calculated using the random-effects model as an effect size to assess variations between studies in addition to sampling errors within studies. For statistical analysis, the Mantel–Haenszel method was used. The *I*^2^ statistic was used to assess heterogeneity. Strong, moderate, and no heterogeneity were indicated by *I*^2^ values of >50, 25–50, and <25%, respectively [24].

## 3. Results

### 3.1. Studies Retrieved and Characteristics

Our initial search returned 1643 studies. After we screened titles and abstracts, 254 studies qualified for a full review (Figure 1). We finally included 25 studies [6,20,25,26,27,28,29,30,31,32,33,34,35,36,37,38,39,40,41,42,43,44,45,46,47] featuring 2554 patients for meta-analysis. Ten studies (*n* = 1289) only reported efficacy outcomes [25,26,27,28,29,30,31,32,33,34], nine studies (*n* = 826) only reported safety outcomes [6,35,36,37,38,39,40,41,42], and six studies (*n* = 439) reported both efficacy and safety outcomes [20,43,44,45,46,47].

Table 1 provides the key characteristics of the included studies. Five studies were prospective observational studies [26,34,37,43,47], and 20 were retrospective observational studies [6,20,25,27,28,29,30,31,32,33,35,36,38,39,40,41,42,44,45,46]. There were no randomized control trials. All studies included adult populations, with most patients having hematologic disorders or a history of solid organ transplantation, and 13 studies were conducted in Asia [6,20,31,36,37,39,40,41,42,44,45,46,47]. Five studies included only patients diagnosed with proven or probable IFIs [25,28,29,43,46], and four studies included only patients who received voriconazole for the targeted/preemptive treatment of invasive aspergillosis [29,31,34,43]. Nine studies included patients who concomitantly used other antifungals [6,25,28,29,30,32,34,43,44]. A summary of outcomes for each study is presented in Table 2. The definitions of efficacy and safety outcomes were not identical across studies.

### 3.2. Risk of Bias

The results of the risk of bias assessment for each study included in the meta-analysis are presented in Figure 2. Regarding the selection of participants, 24 studies were judged as having a low risk of bias, with the remaining study having a high risk of bias because of unclear confirmation of the study site. The risks of selection biases associated with confounding variables were all judged to be high mainly because adequate adjustment was not performed for major confounding variables (age, pathological condition, fungal strains, therapy duration of voriconazole, combined therapy, and targeted/prophylactic therapy). In relation to the measurement of exposure, five studies were judged to have an unclear risk of bias because it was uncertain how the clinical data associated with voriconazole therapy were recorded. The risk of detection biases was considered low for 11 studies and high for 14 studies because no information regarding the blinding methods for assessing efficacy was provided. The risks of attrition biases and reporting biases were deemed low in all but three and four studies, respectively.

### 3.3. Evaluation of Efficacy Outcomes

We set the stepwise cutoffs for the voriconazole trough concentration within a range of 0.5–2.0 μg/mL, and we compared the efficacy rates for values above and below the cutoffs. The random-effects model analysis illustrated that all-cause mortality rates against IFIs were significantly decreased at concentrations of ≥1.0 μg/mL (OR = 0.34, 95% CI = 0.15–0.80, *p* = 0.01, Figure 3).

The overall incidence rate of all-cause mortality at this threshold was 17.7%, compared to 35.1% at concentrations of <1.0 μg/mL. Subgroup analysis revealed that the risk of all-cause mortality was significantly decreased at concentrations of ≥1.0 μg/mL in the following subgroups: Asian study locations (OR = 0.17, 95% CI = 0.06–0.51, *p* = 0.002), prospective observational studies (OR = 0.17, 95% CI = 0.06–0.51, *p* = 0.002), and fungal organisms with *Aspergillus* populations of <100% (OR = 0.34, 95% CI = 0.15–0.80, *p* = 0.01, Figure 4). There was no significant difference at this threshold in other subgroup analyses despite the lower rate of all-cause mortality than in the control groups.

Concerning a voriconazole trough concentration of 0.5 μg/mL, we were unable to pool data because only one study [25] reported all-cause mortality. Although two studies contributed data for all-cause mortality at a concentration of 2.0 μg/mL, our meta-analysis illustrated that its rate of occurrence did not significantly differ according to the cutoff (OR = 0.21, 95% CI = 0.03–1.44, *p* = 0.11, Figure 5).

For treatment success as presented in Figure 6, the meta-analysis demonstrated a significant increase in the rate at voriconazole trough concentrations of ≥1.0 μg/mL (OR = 3.35, 95% CI = 1.52–7.38, *p* = 0.003).

The overall incidence rate of successful treatment at this threshold was 72.0%, compared to 56.3% at concentrations of <1.0 μg/mL. Among the reported studies using a cutoff of 0.5 μg/mL, there was a significant difference in the incidence of treatment success between concentrations above and below the cutoff (OR = 3.48, 95% CI = 1.45–8.34, *p* = 0.005, Figure 7). However, there was no significant difference at a cutoff of 2.0 μg/mL. The results of subgroup analyses for treatment success at each evaluated cutoff are summarized in Appendix A.

### 3.4. Evaluation of Safety Outcomes

We set the stepwise cutoffs for the voriconazole trough concentration within a range of 3.0–6.0 μg/mL, and we compared the safety rates for values above and below the cutoffs. Concerning hepatotoxicity (Figure 8), the meta-analysis illustrated that its rate of occurrence was significantly higher at concentrations of ≥3.0 (OR = 5.66, 95% CI = 3.21–9.99, *p* < 0.001), ≥4.0 (OR = 7.39, 95% CI = 3.81–14.36, *p* < 0.001), ≥5.0 (OR = 5.54, 95% CI = 3.07–9.99, *p* < 0.001), and ≥6.0 μg/mL (OR = 3.71, 95% CI = 2.10–6.55, *p* < 0.001). The overall incidence rate of hepatotoxicity increased as the voriconazole trough concentration increased, with the ORs markedly increasing at concentrations of ≥4.0 μg/mL. Subgroup analysis revealed significant differences for the Asian study locations and for the retrospective observational studies at all cutoffs and for the prospective observational studies at concentrations of ≥5.0 μg/mL (Appendix A).

Concerning neurotoxicity (Figure 9), the meta-analysis illustrated that its rate of occurrence was significantly increased at concentrations of ≥3.0 (OR = 2.64, 95% CI = 1.43–4.86, *p* = 0.002), ≥4.0 (OR = 5.76, 95% CI = 3.14–10.57, *p* < 0.001), ≥5.0 (OR = 5.02, 95% CI = 1.30–19.34, *p* = 0.02), and ≥6.0 μg/mL (OR = 3.67, 95% CI = 1.87–7.18, *p* < 0.001).

Similar to the results for hepatotoxicity, the overall incidence rate of neurotoxicity increased as the voriconazole trough concentration increased, with the ORs markedly increasing at concentrations of ≥4.0 μg/mL. Subgroup analysis revealed significant differences for the Asian study locations at cutoffs of 3.0, 4.0 and 5.0 μg/mL, for the Asian study locations at cutoffs of 4.0–5.0 μg/mL, and for the retrospective observational studies at all cutoffs (Appendix A).

## 4. Discussion

We performed a systematic review and meta-analysis to evaluate the relationships of voriconazole target trough concentrations with efficacy and safety, and the analysis included 2554 adult patients from 25 observational studies. The most important finding of this study was that voriconazole trough concentrations of ≥1.0 μg/mL significantly decreased the all-cause mortality rate in adults with IFIs. Because the voriconazole dosage and frequency were unknown or they varied widely, the dosing schedule was not apparent in this study. However, the optimal trough concentration of voriconazole in adults should be set using a cutoff of ≥1.0 μg/mL based on the result of the TDM-based dose adjustment. This breakpoint is consistent with the results for the treatment success rate, which was the other efficacy outcome in this study.

We further explored the optimal cutoff for efficacy over a range of 0.5–2.0 μg/mL as recommended in some publications [15,16,17,18,19,20]. As a result, although a trough concentration of ≥0.5 or 2.0 μg/mL was not significantly associated with mortality rate, a trough concentration of ≥0.5 μg/mL was significantly associated with treatment success rate. This discrepancy might be because of the low number of studies available for the analysis. Additionally, the reasons may be complex and may include immunological or physiological differences such as age, underlying disease, type of fungal infection, and duration of therapy. However, our subgroup analysis according to study location or design, diagnosis of fungal infection, fungal organisms, and concomitant antifungals for patients with IFIs further validated the importance of a trough concentration of 1.0 μg/mL for efficacy. Meanwhile, previous reports did not find any relationship between voriconazole concentration and survival because the sample size was small (only two studies reported to date), unlike our results [13,14]. Therefore, weak or moderate recommendations have been provided to date regarding the optimal concentration of voriconazole in the national guidelines [1,9,10]. We believe that the evidence from our updated meta-analysis more strongly suggests that a trough concentration of ≥1.0 μg/mL provides optimal clinical success during voriconazole therapy in adult patients with IFIs and reinforces the importance of TDM.

We also performed this meta-analysis to explore the relationship between trough concentrations and safety outcomes through prespecified trough concentration of 3.0–6.0 μg/mL. The hepatotoxicity incidence rates significantly increased as the trough concentration increased, and increased risk was observed at concentrations of ≥3.0 μg/mL, which were considerably lower than those described in previous studies [2,10,14,48]. Similar to the data for hepatotoxicity, we also observed that higher voriconazole concentrations were associated with a greater incidence of neurotoxicity. Our results are in line with those previously reported by Jin et al., in which the incidence rate of hepatotoxicity increased at voriconazole concentrations of ≥3.0 μg/mL [13]. However, they could not find a relationship between neurotoxicity and concentrations of ≥3.0 μg/mL. Additionally, voriconazole concentrations and dosage are highly variable between adults and children because of their different pharmacokinetic profiles [49,50,51,52], although previous meta-analyses did not differentiate between these populations [13,14]. Thus, it is difficult to appreciate the accuracy and validity of their results. Our meta-analysis included many recent studies of only adult patients and provided an in-depth analysis of important variables that might affect the measured outcomes. Furthermore, in our study, the ORs of hepatotoxicity and neurotoxicity markedly increased, especially at trough concentrations of ≥4.0 μg/mL. In a recent multicenter study, Hamada et al. demonstrated that voriconazole-induced hepatotoxicity was improved in almost all patients with dose adjustment based on the initial trough concentration using TDM, and their voriconazole therapy was continued [6]. They also reported that dose adjustment arising from the initial TDM results could not be conducted in approximately 80% of patients with visual symptoms because of the comparatively early onset, but the symptoms were alleviated or eliminated despite continued therapy in most patients. In their study, the median period between the start of therapy and that of TDM was six days (interquartile range = 5–7), and the voriconazole trough concentration cutoffs for predicting adverse effects were 3.5 μg/mL for hepatotoxicity and 4.2 μg/mL for visual symptoms. These results may support the appropriateness of setting the target trough concentration cutoff at <4.0 μg/mL if the opportunity to optimize the voriconazole concentration early is provided based on the initial TDM. The pharmacokinetics of voriconazole are highly variable because of drug–drug interactions, CYP2C19 (and to a lesser extent CYP3A4 and CYP2C9) genetic polymorphisms, and physiological conditions associated with underlying diseases [17,38,53]. Therefore, if the prevention of voriconazole adverse events is important, then we recommend that the optimal trough concentration of voriconazole should be set at <3.0 μg/mL. Conversely, in institutions in which monitoring of voriconazole concentrations has become routine and TDM has been incorporated into the AFS program, we believe that a slightly relaxed target trough concentration of <4.0 μg/mL is a reasonable recommendation to establish a clinical and practical therapeutic range.

This study had several limitations. First, because of the absence of evidence from randomized studies, our conclusions were based only on evidence from observational studies. Second, in terms of design, there was poor reporting of the dosage, methods of administration, assay of trough concentration, definitions of outcomes, and CYP2C19 genotype status. Furthermore, the definitions of efficacy and safety outcomes were not identical across studies. Particularly, visual disturbances are not usually equal to visual hallucinations and more serious neurological adverse events; however, in most studies, they are summarized as neurological adverse events. Thus, inherent biases because of confounding and shortcomings of these studies may have affected our findings. Moreover, publication bias, that is, the possibility that papers that demonstrate an effect of monitoring strategy differences on the primary outcome (all-cause mortality) are preferentially selected and published, was quite likely. However, there was no strong heterogeneity associated with the primary outcome, supporting the consistency of the results. Third, the trials included in this study used several different definitions of efficacy and/or safety outcomes. Thus, to address these issues, future research efforts should involve large-scale prospective randomized clinical trials, which will enable further high-quality meta-analyses.

## 5. Conclusions

A target trough concentration of ≥1.0 μg/mL is likely to be associated with a better survival benefit than that of <1.0 μg/mL in adult patients with IFIs. High trough concentrations significantly increase the risk of hepatotoxicity and neurotoxicity; hence, <4.0 μg/mL has been suggested to be the upper limit of the target trough concentration to minimize toxicity, especially in institutions in which monitoring of voriconazole concentrations had been routinely performed. Additionally, when the prevention of voriconazole-related adverse events is important, a target concentration of <3.0 μg/mL might result in greater potential to reduce the incidence of voriconazole-induced toxicity. Because most evidence was obtained from observational studies, further high-quality trials exploring monitoring strategies for voriconazole use and the effectiveness and safety of voriconazole are needed to expand the research horizons in this area.

## Figures and Tables

**Figure 1 jof-07-00306-f001:**
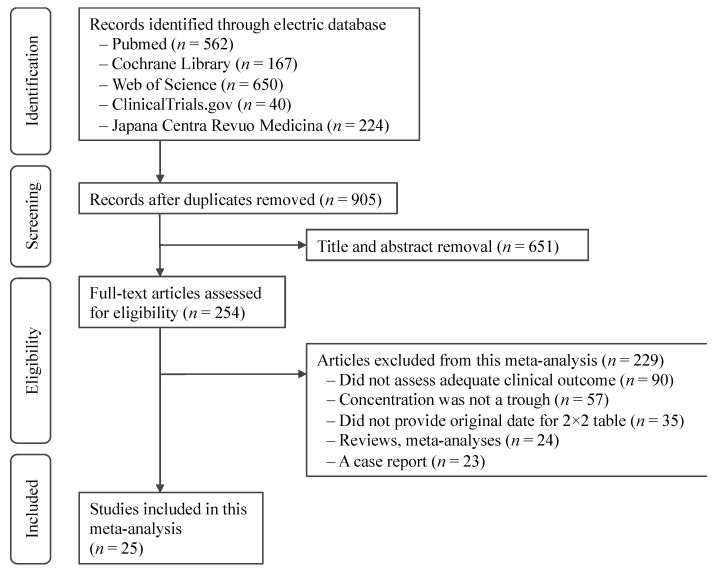
Flowchart of the selection process for the included studies.

**Figure 2 jof-07-00306-f002:**
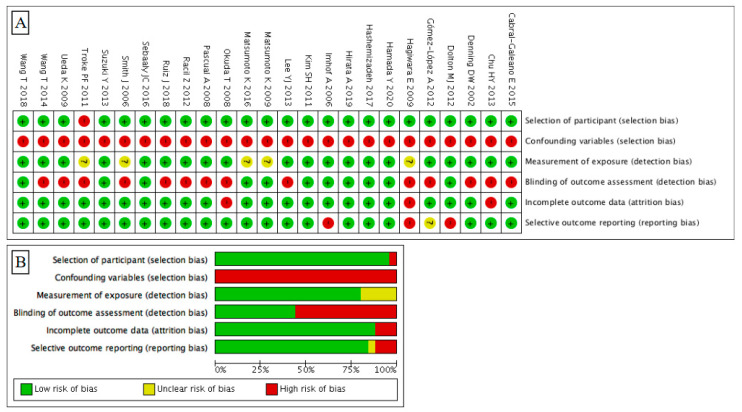
Risk of bias of each study included in this meta-analysis. (**A**) Risk of bias summary: review authors’ judgments about each risk of bias item for each included study. (**B**) Risk of bias graph: review authors’ judgments about each risk of bias item presented as percentages across all included studies. ?: unclear risk of bias; −: high risk of bias; +: low risk of bias.

**Figure 3 jof-07-00306-f003:**
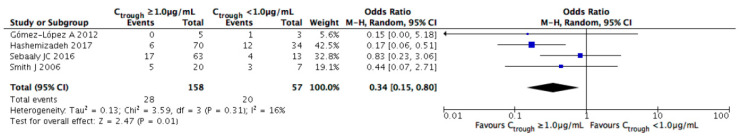
Meta-analysis of all-cause mortality among patients with voriconazole trough concentrations of ≥1.0 μg/mL compared with patients with concentrations of <1.0 μg/mL. The vertical line indicates no significant difference between the groups compared. Diamond shapes and horizontal lines represent odds ratios and 95% confidence intervals (CIs,) respectively. Squares indicate point estimates, and the size of each square indicates the weight of each study included in this meta-analysis. M-H, Mantel–Haenszel; random, random-effects.

**Figure 4 jof-07-00306-f004:**
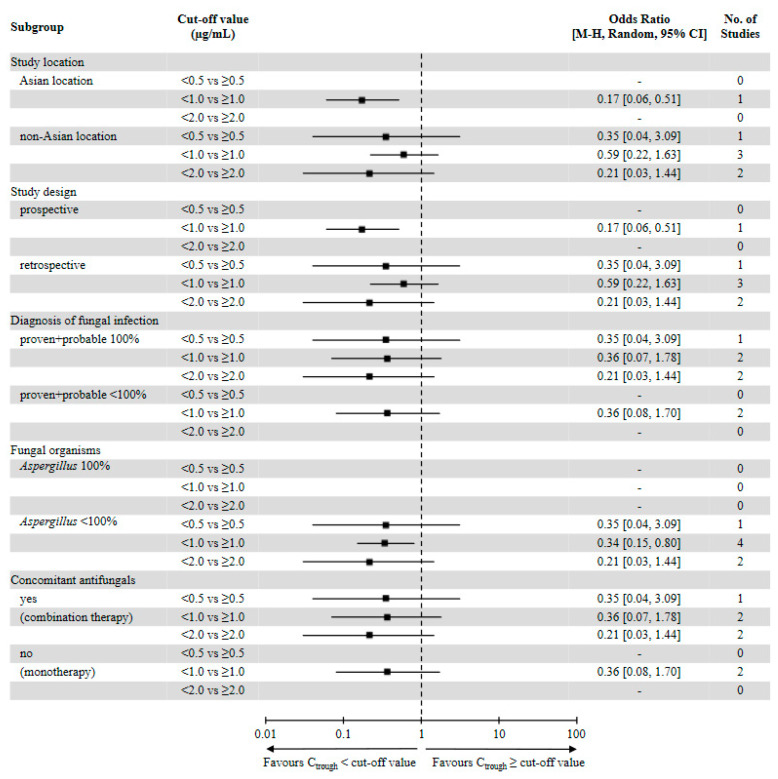
Summary of subgroup analyses for all-cause mortality. Data are presented as forest plots of the odds ratios (ORs) for all-cause mortality according to prespecified baseline subgroups. ORs and 95% confidence intervals (CIs) were calculated using the Mantel–Haenszel (M-H) method according to the random-effects (random) model.

**Figure 5 jof-07-00306-f005:**
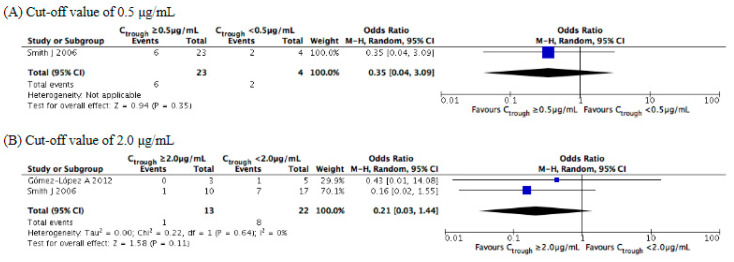
Meta-analysis of the incidence of all-cause mortality at cutoffs of 0.5 and 2.0 μg/mL. The vertical line indicates no significant difference between the groups compared. Diamond shapes and horizontal lines represent odds ratios and 95% confidence intervals (CIs), respectively. Squares indicate point estimates, and the size of each square indicates the weight of each study included in this meta-analysis. M-H, Mantel–Haenszel; random, random-effects.

**Figure 6 jof-07-00306-f006:**
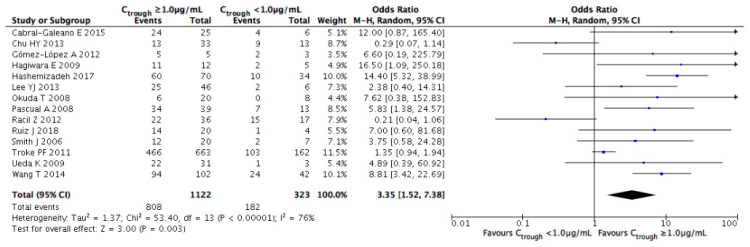
Meta-analysis of treatment success among patients with voriconazole trough concentrations of ≥1.0 μg/mL compared with patients with concentrations of <1.0 μg/mL. The vertical line indicates no significant difference between the groups compared. Diamond shapes and horizontal lines represent odds ratios and 95% confidence intervals (CIs), respectively. Squares indicate point estimates, and the size of each square indicates the weight of each study included in this meta-analysis. M-H, Mantel–Haenszel; random, random-effects.

**Figure 7 jof-07-00306-f007:**
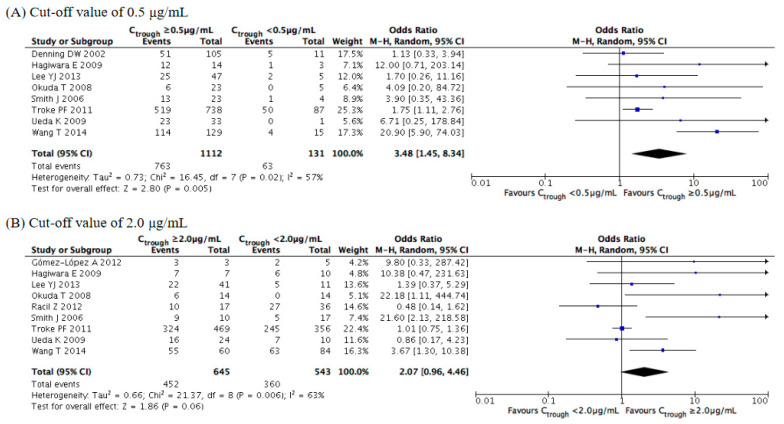
Meta-analysis of the incidence of treatment success at cutoffs of 0.5 and 2.0 μg/mL. The vertical line indicates no significant difference between the groups compared. Diamond shapes and horizontal lines represent odds ratios and 95% confidence intervals (CIs), respectively. Squares indicate point estimates, and the size of each square indicates the weight of each study included in this meta-analysis. M-H, Mantel–Haenszel; random, random-effects.

**Figure 8 jof-07-00306-f008:**
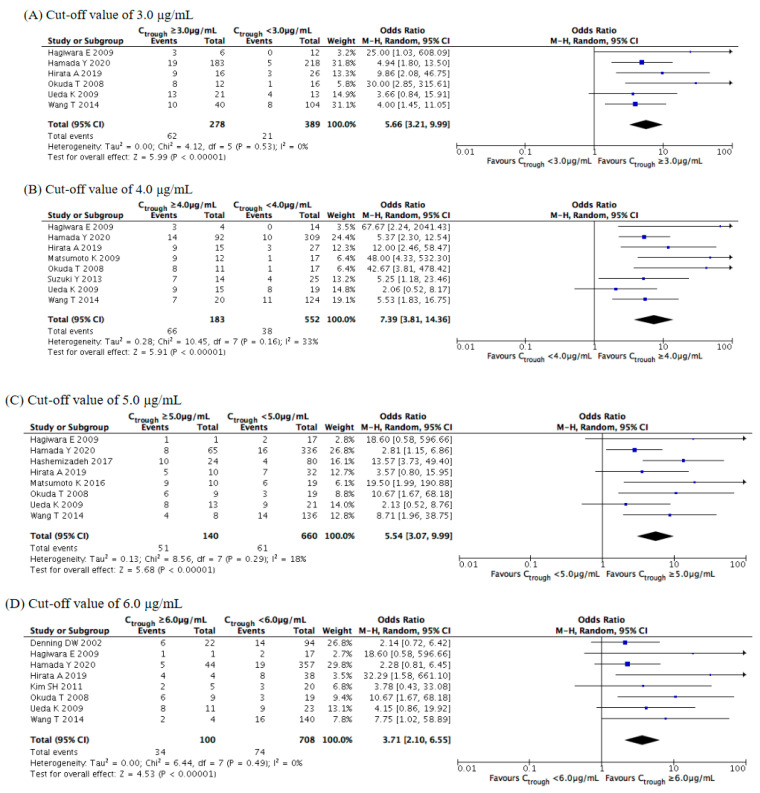
Meta-analysis of the incidence of hepatotoxicity at cutoffs of 3.0–6.0 μg/mL. The vertical line indicates no significant difference between the groups compared. Diamond shapes and horizontal lines represent odds ratios and 95% confidence intervals (CIs), respectively. Squares indicate point estimates, and the size of each square indicates the weight of each study included in this meta-analysis. M-H, Mantel–Haenszel; random, random-effects.

**Figure 9 jof-07-00306-f009:**
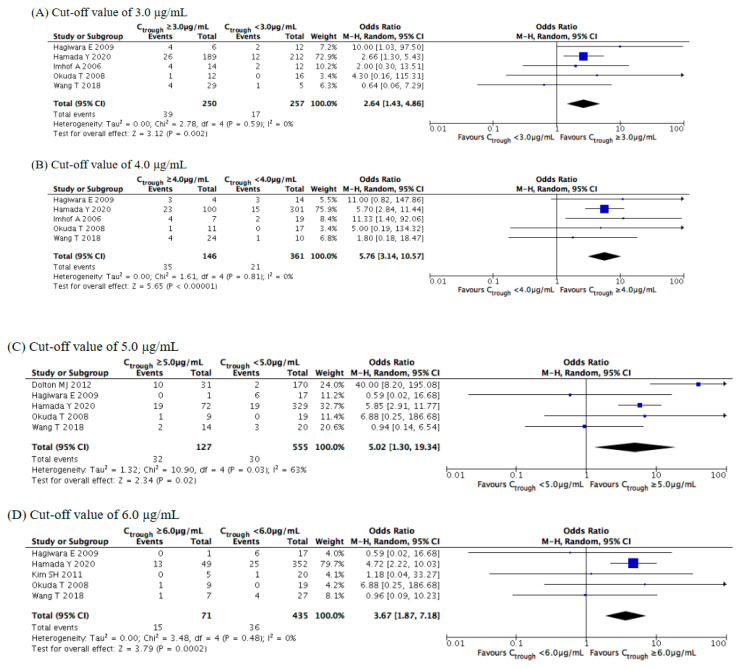
Meta-analysis of the incidence of neurotoxicity at cutoffs of 3.0–6.0 μg/mL. The vertical line indicates no significant difference between the groups compared. Diamond shapes and horizontal lines represent odds ratios and 95% confidence intervals (CIs), respectively. Squares indicate point estimates, and the size of each square indicates the weight of each study included in this meta-analysis. M-H, Mantel–Haenszel; random, random-effects.

**Table 1 jof-07-00306-t001:** Characteristics of the studies included in the meta-analysis.

Study	Year	Study Location	Study Design	Number of Cases	Age (Years)	Main Underlying Disease (%)	Indication of Therapy	Diagnosis of Fungal Infection (%)	Fungal Organisms (%)	Duration of Therapy (Days)	Concomitant Antifungals
Denning DW	2002	UK	MCP	116	median: 52 range: 18–79	haematological disorder (58) allogeneic HSCT (20)	targeted	proven (41) probable (59)	*Aspergillus* (100)	NR	yes
Smith J ^a^	2006	USA	SCR	27	median: 40 range: 16–70	solid organ transplantation (59) bone marrow transplantation (19)	targeted	proven or probable (100)	*Aspergillus* (85) *Candida* (4)	NR	yes
Imhof A	2006	Switzerland	SCR	26	median: 47.5 range: 22–61	acute myeloid leukaemia (89)	targeted	proven (27) probable (19) possible (54)	*Aspergillus* or *Candida* (100)	NR	no
Pascual A	2008	Switzerland	SCP	52	median: 58.5 range: 23–78	haematological malignancy (61) solid organ transplantation (6)	targeted	proven or probable (69) possible (21)	*Aspergillus* (50) *Candida* (15)	median: 50 range: 4–1130	no
Okuda T ^b^	2008	Japan	SCR	23	median: 64 range: 18–85	myelodysplastic syndrome (35) acute myeloid leukaemia (17) acute lymphatic leukaemia (9)	targeted	proven or probable (65)	*Aspergillus* (48)	NR	yes
Ueda K	2009	Japan	SCR	34	median: 57.5 range: 19–81	acute myeloid leukaemia (56) non-hodgkin lymphoma (24)	targeted	proven (2) probable (15) possible (83)	NR	NR	no
Hagiwara E ^c^	2009	Japan	SCR	18	median: 67 range: 53–80	respiratory disease (100)	targeted	proven (33) probable or possible (67)	*Aspergillus* (33)	NR	no
Matsumoto K	2009	Japan	SCR	29	mean: 57.3 SD: ±19.3	NR	targeted	NR	NR	NR	no
Troke PF	2011	UK	MCR	825	median: 44	haematological malignancy	targeted	NR	NR	NR	no
Kim SH	2011	Korea	SCP	25	median: 45 IQR: 38–54	acute myeloid leukaemia (56) acute lymphatic leukaemia (24)	targeted	NR	NR	median: 8 IQR: 7–14	no
Gómez-López A ^d^	2012	Spain	SCR	8	median: 70 range: 17–75	haematological malignancy (63) solid organ transplantation (13)	targeted	proven (37) probable (63)	*Aspergillus* (63)	median: 33 range: 9–243	yes
Racil Z ^e^	2012	Czech Republic	MCR	53	range: 18–77	NR	targeted	proven (21) probable (79)	*Aspergillus* (100)	median: 32 range: 5–160	yes
Dolton MJ	2012	Australia	SCR	201	median: 54 range: 18–88	haematological malignancy (59) solid organ transplantation (13)	targeted or prophylactic	proven (22) probable (11) possible (29)	*Aspergillus* (19) *Candida* (5)	NR	no
Chu HY	2013	USA	SCR	108	median: 53 IQR: 38–64	HSCT (44) haematological malignancy (34) solid organ transplantation (9)	targeted	proven (7) probable (36) possible (40)	*Aspergillus* (81) *Candida* (8)	median: 35 range: 13–92	yes
Lee YJ	2013	Korea	SCR	52	range: 16–81	acute myeloid leukaemia (60) acute lymphatic leukaemia (13) myelodysplastic syndrome (10)	targeted	proven (4) probable (56) possible (40)	*Aspergillus* (100)	range: 23–131	no
Suzuki Y	2013	Japan	SCR	39	mean: 55.9 SD: ±19.5	NR	NR	NR	NR	mean: 58.4 SD: ±32.0	no
Wang T	2014	China	SCR	144	median: 60.6 range: 18–99	bronchitis (24) liver diseae (22) asthma (19) haematological malignancy (15) solid organ transplantation (7)	targeted	proven (61) probable (39)	*Aspergillus* (62) *Candida* (32)	median: 35 range: 16–81	no
Cabral-Galeano E ^f^	2015	Spain	SCR	52	median: 55 IQR: 37.5–60.7	lung transplantation (48) haematological disorder (27) cystic fibrosis (10)	targeted or prophylactic	proven (90)	NR	median: 8 IQR: 3–14	yes
Sebaaly JC ^g^	2016	USA	SCR	88	mean: 52.7 SD: ±14.8	haematological malignancy (40) solid organ transplantation (23) stem cell transplantation (17) pulmonary disease (15)	targeted	NR	NR	median: 18 IQR: 11–26	no
Matsumoto K	2016	Japan	SCR	29	mean: 58.6 range: 16–77	haematological malignancy (52) solid tumour (21) respiratory disease (14)	NR	proven (59) probable or possible (21)	*Aspergillus* (59) *Candida* (14)	median: 22 range: 6–171	no
Hashemizadeh Z ^h^	2017	Iran	SCP	104	mean: 36 range: 18–62	solid organ transplantation (100)	targeted	proven (8) probable (40) possible (52)	*Aspergillus* (40) *Candida* (8)	mean: 54 range: 29–98	no
Ruiz J	2018	Spain	SCP	24	mean: 55.3 SD: ±12.6	NR	targeted	proven or probable (75) possible (25)	*Aspergillus* (100)	NR	yes
Wang T	2018	China	MCR	34	range: 22–82	NR	NR	NR	NR	NR	no
Hirata A	2019	Japan	SCR	42	mean: 61.9 SD: ±16.9	haematological malignancy (90)	targeted or prophylactic	proven (40) probable or possible (33)	*Aspergillus* (36) *Candida* (5)	NR	no
Hamada Y ^i^	2020	Japan	MCR	401	mean: 61.8 range: 18–91	haematological malignancy (39) collagen disease (22) solid organ malignancy (10)	targeted or prophylactic	proven (25) probable or possible (52)	*Aspergillus* (9) *Candida* (11)	median: 30 IQR: 14–117	yes

NR, not reported; SCR, single-center retrospective; SCP, single-center prospective; MCR, multi-center retrospective; MCP, multi-center prospective; IQR, interquartile range; SD, standard deviation; HSCT, hematopoietic stem cell transplant. ^a^ One pediatric patient (age < 15) was excluded from the analysis. ^b^ Twenty-eight voriconazole concentrations obtained from 23 patients were analyzed in the study. ^c^ Seventeen patients were considered assessable for efficacy, and 18 patients were considered assessable for safety. ^d^ Four pediatric patients (age < 15) and two patients without a voriconazole concentration measurement were excluded from the analysis. ^e^ The subgroup diagnosed with invasive aspergillosis was used. ^f^ Thirty-one patients were considered assessable for efficacy. ^g^ Twelve patients who used voriconazole for prophylaxis were excluded from the analysis. ^h^ Concentrations of >1.3 μg/mL were set as 1.0 μg/mL for efficacy, and those of >5.3 μg/mL were set as 5.0 μg/mL for safety. ^i^ Obtained additional data from the authors.

**Table 2 jof-07-00306-t002:** Summary of outcomes and definitions of included studies.

Scheme	Year	Reported Outcome	Definition of Treatment Success	Definition of Hepatotoxicity	Definition of Neurotoxicity
Denning DW	2002	treatment success hepatotoxicity	complete, partial or stable response based on clinical and radiological evidence	AST or ALT > 5 times ULN, ALP > 3 times ULN, TBIL > 3 times ULN	-
Smith J	2006	treatment success all-cause mortality	absence of progression of lesions on follow-up imaging	-	-
Imhof A	2006	neurotoxicity	-	-	neuropathy, hallucinations, confusion, anxiety, asthenia, visual disturbance, dysarthria or insomnia
Pascual A	2008	treatment success	absence of persistence or progression of fungal infection (based on clinical and radiological evidence) and proven or persumed eradication of the fungal pathogen	-	-
Okuda T	2008	treatment success hepatotoxicity neurotoxicity	*β*-D-glucan or *Aspergillus* antigen decrease below standard at follow-up	any deviation in AST or ALT from the normal range	hallucination
Ueda K	2009	treatment success hepatotoxicity	absence of at least 2 of the following 3 types of criteria: clinical (development of new fever or persistent fever), radiologic (>25% expansion of abnormal shadow area in CT image), or mycological (a rise within the abnormal range or positive conversion of serum markers (either *β*-D-glucan or galactomannan)) failure	AST, ALT, GGT or TBIL was in gredes 2–4 according to NCI criteria	-
Hagiwara E	2009	treatment success hepatotoxicity neurotoxicity	complete or partial response based on clinical and radiologic evidence	liver function test > 3 times ULN	any visual symptoms
Matsumoto K	2009	hepatotoxicity	-	AST, ALT, ALP, GGT or TBIL was in gredes 1–4 according to NCI criteria	-
Troke PF	2011	treatment success	complete or partial response based on EORTC/MSG criteria	-	-
Kim SH	2011	hepatotoxicity neurotoxicity	-	NCI, grades 3–5 was referred to as severe adverse events	NCI, grade 3–5
Gómez-López A	2012	treatment success all-cause mortality	complete or partial response based on EORTC/MSG criteria	-	-
Racil Z	2012	treatment success	complete or partial response based on EORTC/MSG criteria	-	-
Dolton MJ	2012	neurotoxicity	-	-	visual/auditory hallucinations
Chu HY	2013	treatment success	complete or partial response based on clinical, radiologic and microbiologic evidence	-	-
Lee YJ	2013	treatment success	complete or partial response based on EORTC/MSG criteria	-	-
Suzuki Y	2013	hepatotoxicity	-	AST, ALT, ALP, GGT or TBIL was in gredes 2–4 according to NCI criteria	-
Wang T	2014	treatment success hepatotoxicity	absence of persistence or progression of fungal infection (based on clinical and radiological evidence) and proven or persumed eradication of the fungal pathogen	AST, ALT, ALP or TBIL was in gredes 3–4 according to CTCAE criteria	-
Cabral-Galeano E	2015	treatment success	absence of persistence or progression of fungal infection based on clinical and radiological evidence	-	-
Sebaaly JC	2016	all-cause mortality	-	-	-
Matsumoto K	2016	hepatotoxicity	-	AST, ALT, ALP, GGT or TBIL was in gredes 1–4 according to CTCAE criteria	-
Hashemizadeh Z	2017	treatment success all-cause mortality hepatotoxicity	complete or partial response based on EORTC/MSG criteria	AST, ALT, ALP or TBIL was in gredes 2–4 according to CTCAE criteria	-
Ruiz J	2018	treatment success	complete or partial response based on clinical and radiologic evidence	-	-
Wang T	2018	neurotoxicity	-	-	dizziness, tremor, hallucinations, encephalopathy, consciousness disturbance
Hirata A	2019	hepatotoxicity	-	AST, ALT, ALP or TBIL was in gredes 2–4 according to CTCAE criteria	-
Hamada Y	2020	hepatotoxicity neurotoxicity	-	AST or ALT > 3 times ULN or >3 times the baseline if AST or ALT baseline was abnormal	any visual symptoms (colour perception, blurred vision, bright spots, wavy lines and photophobia)

EORTC/MSG, Mycoses Study Group and European Organization for Research and Treatment of Cancer; AST, aspartate aminotransferase; ALT, alanine aminotransferase; ALP, alkaline phosphatase; TBIL, total bilirubin; ULN, upper limit of normal; NCI, National Cancer Institute; CTCAE, Common Terminology Criteria for Adverse Events.

## Data Availability

The data presented in this study are available on request from the corresponding author.

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
