# Peer review of "Favorable Effects of Voriconazole Trough Concentrations Exceeding 1 μg/mL on Treatment Success and All-Cause Mortality: A Systematic Review and Meta-Analysis"

_jof, 2021, doi:10.3390/jof7040306_

Round 1

Reviewer 1 Report

Overall, I think the meta-analysis has been done very thoroughly and methodologically correct. All relevant papers till now have been included. Antifungal stewardship is a hot topic these days and the manuscript underscores once more (and very convincingly)that a through level of 1 ng/ml should be targeted to improve survival but it also provides new thresholds for avoiding toxicity.

Author Response

Reviewer 1: Comments and Suggestions for Authors

Overall, I think the meta-analysis has been done very thoroughly and methodologically correct. All relevant papers till now have been included. Antifungal stewardship is a hot topic these days and the manuscript underscores once more (and very convincingly)that a through level of 1 ng/ml should be targeted to improve survival but it also provides new thresholds for avoiding toxicity.

Response: Thank you for your kind comments.

Reviewer 2 Report

The manuscript submitted by Hanai and colleagues reports the results of a systematic review and metaanalysis examining the optimal through target concentration associated with treatment success in adult patients with invasive fungal diseases (IFDs).  The analysis included 25 studies involving 2554 patients. The probability of mortality was significantly reduced using a cutoff of ≥1.0 μg/mL (odds ratio [OR] = 0.34); cutoffs of 0.5 (OR = 3.48, 95% CI = 1.45–8.34) and 1.0 μg/mL 22 (OR = 3.35, 95% CI = 1.52–7.38) increased the treatment success rate. Significantly higher risks of hepatotoxicity and neurotoxicity were demonstrated at higher concentrations for all cutoffs, and the highest ORs were recorded at 4.0 μg/mL (OR = 7.39).

Altogether, this is an interesting, timely, and well-written contribution that confirms known associations between exposure of voriconazole and treatment success and occurrence of adverse events in the setting of treatment of IFDs.

I have the following comments and/or suggestions:

  1. Introduction: Well written and focused on the rationale and formulation of the question analyzed.
  2. Methods: Pl. clarify whether only data related on treatment of proven or probable IFDs were included, or provide the criteria accepted for diagnosis of IFDs. If the data analyzed included confirmed and suspected IFDs, you should state this throughout.
  3. Methods: Pl. provide a reference for definition of treatment success or provide a more in-depth description.
  4. Methods: For completeness of the review process and not for reasons of doubts on its validity, I suggest review of the statistical methodology by a statistician.
  5. Results: Well-structured, providing appropriate and objective information on the analyses and their results.
  6. Results: The definitions of hepatotoxicity and of neurotoxicity in the individual studies need consideration in the discussion.
  7. Discussion: The authors analyzed data from adults. Is there any reason to not translate the thresholds obtained in this analysis to pediatric patients? Pl. comment.
  8. Discussion: In the reviewer’s opinion, visual disturbances are not equal to visual hallucinations and more serious neurological AEs; however, in most studies, they are summarized as neurological AEs. While this cannot be changes by the authors, a brief discussion of this point may be appropriate. Please also see comment 6 and include this point in the list of limitations.
  9. Discussion: Since voriconazole may cause induction of its own hepatic metabolism (autoinduction), the point of repeat TDM time points during a course of treatment is important in your TDM recommendations.
  10. Figures and tables: Appropriate, providing complete information of study characteristics and analysis outcomes.

Author Response

Reviewer 2: Comments and Suggestions for Authors

Response: Thank you for your kind and important comments.

The comment of Reviewer #2

Response

1.    Introduction: Well written and focused on the rationale and formulation of the question analyzed.

Thank you for your comments.

2.    Methods: Pl. clarify whether only data related on treatment of proven or probable IFDs were included, or provide the criteria accepted for diagnosis of IFDs. If the data analyzed included confirmed and suspected IFDs, you should state this throughout.

Line 120-121: The efficacy outcomes were all-cause mortality and treatment success related on treatment of confirmed or suspected IFIs.

3.    Methods: Pl. provide a reference for definition of treatment success or provide a more in-depth description.

Thank you for your important point. We added the information about definition of treatment success in the Table 2 (summary of outcomes and definitions of included studies).

4.    Methods: For completeness of the review process and not for reasons of doubts on its validity, I suggest review of the statistical methodology by a statistician.

Unfortunately, we could not have the opportunity to have our paper reviewed by a statistician; however, we consider that the meta-analysis has been done methodologically correct. The reason is that the authors conducted a lot of meta-analysis. There are many papers as a result. The study was also checked by two co-authors, professional doctors.

5.    Results: Well-structured, providing appropriate and objective information on the analyses and their results.

Thank you for your comments.

6.    Results: The definitions of hepatotoxicity and of neurotoxicity in the individual studies need consideration in the discussion.

We consider that the heterogeneity of the definition of hepatotoxicity and of neurotoxicity is a limitation of this study. Therefore, we added following description regarding this point in the limitation section.

1.         Line 358-361: Furthermore, the definitions of efficacy and safety outcomes were not identical across studies. Particularly, visual disturbances are not usually equal to visual hallucinations and more serious neurological adverse events; however, in most studies, they are summarized as neurological adverse events.

7.    Discussion: The authors analyzed data from adults. Is there any reason to not translate the thresholds obtained in this analysis to pediatric patients? Pl. comment.

In children, the pharmacokinetics and pharmacodynamics of voriconazole has not yet been fully understood. However, there is an agreement that children require higher dosages to maintain serum concentrations within the therapeutic range. Therefore, it is unclear whether the thresholds obtained in this analysis can be applied to pediatric patients.

8.    Discussion: In the reviewer’s opinion, visual disturbances are not equal to visual hallucinations and more serious neurological AEs; however, in most studies, they are summarized as neurological AEs. While this cannot be changes by the authors, a brief discussion of this point may be appropriate. Please also see comment 6 and include this point in the list of limitations.

Thank you for the important point. We consider that the heterogeneity of the definition of hepatotoxicity and of neurotoxicity is a limitation of this study. Therefore, we added following description regarding this point in the limitation section.

1.         Line 358-361: Furthermore, the definitions of efficacy and safety outcomes were not identical across studies. Particularly, visual disturbances are not usually equal to visual hallucinations and more serious neurological adverse events; however, in most studies, they are summarized as neurological adverse events.

9.    Discussion: Since voriconazole may cause induction of its own hepatic metabolism (autoinduction), the point of repeat TDM time points during a course of treatment is important in your TDM recommendations.

Thank you for your comments. There are some unclear points about own hepatic metabolism (autoinduction), and this time, it could not be examined by meta-analysis, and I thought it would be appropriate not to describe it in the discussion.

10. Figures and tables: Appropriate, providing complete information of study characteristics and analysis outcomes.

Thank you for your comments.
